# Intergrated Transcriptomic and Proteomic Analysis Revealed the Differential Responses to Novel Duck Reovirus Infection in the Bursa of Fabricius of *Cairna moschata*

**DOI:** 10.3390/v14081615

**Published:** 2022-07-25

**Authors:** Tao Yun, Jionggang Hua, Weicheng Ye, Zheng Ni, Liu Chen, Yinchu Zhu, Cun Zhang

**Affiliations:** State Key Laboratory for Managing Biotic and Chemical Threats to the Quality and Safety of Agro-Products, Institute of Animal Husbandry and Veterinary Sciences, Zhejiang Academy of Agricultural Sciences, Hangzhou 310021, China; huajg@126.com (J.H.); ywc119@aliyun.com (W.Y.); nizheng79@163.com (Z.N.); haoliuzi@126.com (L.C.); zhuyc111@163.com (Y.Z.)

**Keywords:** *Cairna moschata*, differential accumulated protein, reovirus, proteomic analysis, transcriptomic analysis, the bursa of Fabricius

## Abstract

The bursa of Fabricius is an immunologically organ against the invasion of duck reovirus (DRV), which is a fatal bird virus belonging to the Reoviridae family. However, responses of the bursa of Fabricius of Cairna moschata to novel DRV (NDRV) infection are largely unknown. Transcriptomes and proteomes of the samples from control and two NDRV strain (HN10 and JDm10) with different virulence were analyzed. Differentially expressed genes and differential accumulated proteins were enriched in the serine protease system and innate immune response clusters. Most of the immune-related genes were up-regulated under both JDm10/HN10 infections. However, the immune-related proteins were only accumulated under HN10 infection. For the serine protease system, coagulation factor IX, three chains of fibrinogen, and complements C8, C5, and C2s were significantly up-regulated by the HN10 infection, suggesting that the serine protease-mediated immune system might be involved in the resistance to NDRV infection. For the innate and adaptive immune system, RIG-I, MDA5, MAPK20, and IRF3 were significantly up-regulated, indicating their important roles against invaded virus. TLR-3 and IKBKB were only up-regulated in the liver cells, MAPK20 was only up-regulated in the bursa of Fabricius cells, and IRAK2 was only up-regulated in the spleen samples. Coagulation factor IX was increased in the bursa of Fabricius, not in the liver and spleen samples. The data provides a detailed resource for studying the proteins participating in the resistances of the bursa of Fabricius of duck to NDRV infections.

## 1. Introduction

Duck reovirus (DRV) is an aquatic bird virus belonging to the genus Orthoreovirus in the family Reoviridae [1]. Recently, several novel DRVs (NDRV) were isolated in different countries and regions [2,3]. The mortality rate of the NDRV infected ducks reached 50%, and caused many duck species (Muscovy duck, Semi-Muscovy duck, Sheldrake duck, and Pekin duck) and geese disease, and has been a great threat to the poultry industry, particularly the Muscovy duck and meat duck industries in China [2,3,4]. Disease that was caused by the NDRV infection displayed a series of serious symptoms, which were characterized mainly by hemorrhagic-necrotic lesions in various organs [2,3,5]. So far, few commercial vaccines have been developed against NDRV infection [6]. Thus, the regulation mechanism underlying the pathogenesis of NDRV is becoming a research hotspot.

Innate and adaptive immunity is an important host defense system to resist virus infections [7]. The seine protease system, consisting of the complement, coagulation, and fibrinolytic sub-systems, is an essential sentinel of innate immunity [8]. Pattern recognition receptors (PRRs) are involved in the initiation of the innate and adaptive immune system by recognizing pathogen-associated molecular patterns (PAMPs) or danger-associated molecular patterns (DAMPs) [9]. The deeply identified PRRs are toll-like receptors (TLRs), a transmembrane receptor family with leucine-rich-repeat motifs [10]. Addition to TLRs, the RIG-I-like receptors (RLRs), including retinoic acid inducible gene I (RIG-I), melanoma differentiation associated genes 5 (MDA-5), and laboratory of genetics and physiology 2 (LGP2), is another PRR family [11]. Then, TLRs and RLRs activate many kinases and transcription factors to protect the host from virus infections. Many adaptor molecules have been discovered recently [7]. For examples, myeloid differentiation primary response protein 88 (MyD88) is a representative adaptor molecule that is used by most of the TLR family members [12]. The adaptor protein with TIR-domain, such as interferon-β (TRIF) and TRIF-related adaptor molecules (TRAMs), are also known as adaptor molecules associated with TLR signaling [12]. Recent studies showed that DRV rapidly initiates the host innate immune system through regulating the RIG-I, MDA5, and TLR3-dependent signaling pathways. Through adaptor molecules, a number of transcription factors, such as NF-κB factors, AP-1 factor, and IRFs, are activated in the host nucleus [13].

The avian immune system consists mainly of bursa of Fabricius, liver, spleen, and blood [14]. The bursa of Fabricius is an immunologically hollow oval chestnut-like sac located dorsally to the cloaca [15]. In duck, the bursa of Fabricius plays essential roles in sustaining the normal immune function to maintain health [16]. Previous studies have identified many DRV infection responsive proteins in the liver and spleen of duck [17,18]. For example, the responses of toll-like receptor gene to reovirus infection was investigated in Peking ducks [19]. However, the responses to DRV infections in the bursa of Fabricius are largely unknown. With the recent development of high-throughput sequencing technology, integrated transcriptomic and proteomic analysis has become a routine tool to reveal the molecular mechanism of poultry [20]. To date, many DRV strains with different virulent have been identified. However, there are few naturally attenuated strains of NDRV. So far, only a naturally attenuated strain N20 has been isolated in the clinic [21]. In our lab, JDm10 was continuously subcultured in DF-1 cells with 150 generation to produce an attenuated strain JDm10 [22]. No significant clinical symptom can be observed in JDm10-injected ducks. In the present study, a number of DRV (HN10 and JDm10) responsive proteins and genes were investigated in the bursa of Fabricius of ducks. The data will provide valuable information on the pathogenicity of DRV in the bursa of Fabricius of ducks.

## 2. Materials and Methods

### 2.1. Ethics Statement and Sampling Method

All tests were carried out in Zhejiang Academy of Agricultural Sciences (ZAAS). All ducks were treated to comply with the Regulations of the Administration of Affairs Concerning Experimental Animals approved by the State Council of China. The bird regulation applied in the present work was approved by the Research Ethics Committee of ZAAS (permit ID: ZAAS2020016). One-day-old Muscovy ducklings were selected for the virus challenge experiment.

### 2.2. Sampling and Protein Pretreatment

The infection experiment was performed using one-day-old Muscovy ducklings, and the serum samples of each Muscovy duckling were tested by RT-PCR and ELISA to ensure that they were free of NDRV pathogens and antibodies [23]. The virulent strain HN10 was isolated from a hemorrhagic and necrotic Muscovy duck liver, and sub-cultured on DF-1 cells for 20 generations [17,18]. The attenuated strain JDm10, was obtained from the liver of Muscovy duckling with a few white necrotic spots of smaller size and further attenuated by passage in DF-1 cells for 150 generations [22].

Twenty-seven healthy Muscovy ducklings were selected and divided into three random groups. One group (nine ducklings) was inoculated intramuscularly with 500 μL of JDm10 strain at a titer of 10^8.2^ median tissue culture infective dose (TCID50) per mL; another group (nine ducklings) was treated intramuscularly with HN10 strain with a titer of 10^6.4^ TCID50 per mL at the same dose as the JDm10 group; the last group (nine ducklings) was treated with DMEM solution with the same conditions. The bursa of Fabricius samples of ducklings of each group were harvested at 72 h post infection (hpi) and frozen in liquid nitrogen until used.

Samples were ground in a mortar with liquid nitrogen and put into a 5 mL tube. Then, the sample was added with pre-chilled working buffer and sonicated using an ultrasonic processor [18]. The sample solution was precipitated with pre-chilled TCA buffer and extracted with 15,000× *g* centrifugation for 15 min at 4 °C. Finally, the precipitate was washed twice with pre-chilled acetone solution and dissolved in storage TEAB buffer.

### 2.3. Trypsin Digestion

Before trypsin digestion, samples were reduced with 10 mM dithiothreitol at 37 °C for 25 min and alkylated with 20 mM iodoacetamide at 37 °C for 30 min. Promega Trypsin (Madison, WI, USA) was used to produce the digested peptides. The trypsin was applied at a mass ratio of 1:50 for a 24 h digestion and at a mass ratio of 1:100 for 4 h digestion.

### 2.4. Tandem Mass Tags (TMT) Labeling and HPLC-MS/MS Analysis

The digested peptide samples were desalted using a Phenomenex Strata X-C18-SPE system (Torrance, CA, USA) and dried by vacuum centrifugation. The resulting samples were redissolved in 0.5 M triethylammonium bicarbonate solution and TMT labeled using a six-plex TMT kit (Thermo-Scientific, Rockford, IL, USA) in accordance with its mammal. The TMT-labeled peptide samples were reconstituted in acetonitrile solution and were vacuum-dried.

The peptides were fractioned using an Agilent 300Extend C18 column on a high pH reverse-phase HPLC system (Santa Clara, CA, USA). The peptides were first divided into 80 fractions according to the previous work [18]. Each fraction was harvested and centrifugation dried. LC-MS/MS analysis was performed in the same way as in the previous published study [17,18].

### 2.5. Bioinformatic Analysis

The saw MS/MS data was uploaded onto the proteome Exchange Consortium database by the PRIDE partner repository with ID PXD025093. Then, the MS/MS data was searched against the unip_Anas_8839 database using Thermo Proteome Discoverer v.2.1.0.81. The bioinformatic analyses, including Gene Ontology (GO) annotation, Kyoto Encyclopedia of Genes and Genomes (KEGG) annotation, protein domain annotation, subcellular localization, and protein–protein interaction, were performed according to our previous studies [17,18].

### 2.6. PPI Network Construction

To demonstrate the potential PPI network, the differentially accumulated proteins (DAPs) were mapped to the PPI database. The PPI relationships with absolute e Pearson Correlation Coefficient > 0.75 were considered as significant. Then, a PPI network was constructed using the Cystoscope software based on their PPI relationships.

### 2.7. Validation of Differentially Accumulated Proteins (DAPs) by Parallel Reaction Monitoring (PRM)

The DAPs detected in the LC-MS/MS-TMT experiment were confirmed using the PRM assay [18]. Briefly, the protein samples were extracted, and trypsin digested using the same method. The tryptic peptides were dissolved in solution A containing 0.1% formic acid and loaded onto an EASY-nLC UPLC system with a home-made reversed-phase analytical column.

### 2.8. Library Construction and RNA Sequencing

Total RNAs from the same samples were extracted using the TRIzol reagent (Invitrogen, Shanghai, China), and contaminated DNA was removed by application of DNase I. The purity, concentration, and integrity of RNAs were determined by a NanoPhotometer^®^ spectrophotometer, a Qubit^®^ RNA Assay Kit in Qubit^®^ 2.0 Flurometer, and an RNA Nano 6000 Assay Kit of the Bioanalyzer 2100 system (Agilent Technologies, CA, USA), respectively. The cDNA libraries were constructed using the NEBNext^®^ Ultra™ Directional RNA Library Prep Kit (NEB, USA) following the manufacturer’s method. After quality checking, the libraries were uploaded to an Illumina Hiseq 4000 platform, and 300 bp (±50 bp) paired-end reads were generated.

### 2.9. Transcriptomic Analysis

Clean reads were produced from the raw reads by removing adapter sequences, duplicated sequences, and low-quality reads with ambiguous bases (“N” > 5%). The high-quality clean reads were de novo assembled into reference sequence using the Trinity program [24]. All clean reads were mapped onto the reference sequence to form contigs, which were estimated and assembled to form unigenes. All unigenes were checked against several databases, Nr (non-redundant), Swiss-Prot (a manually annotated and reviewed protein sequence database), Pfam (protein family), KEGG (Kyoto Encyclopedia of Genes and Genomes), and COG (Clusters of Ortholog Groups) with a cutoff E value of 0.00001 [25].

The mapped reads of each sample were assembled and quantified by StringTie (v1.3.1) in a reference-based method, http://ccb.jhu.edu/software/stringtie/, accessed on 3 December 2021). Differentially expressed genes (DEGs) were identified by DESeq2 [26]. Genes with a corrected *p* value < 0.01 and fold changes >1.5 or <0.67 were assigned as significant DEGs.

### 2.10. Real-Time PCR

Differential expression of randomly selected key protein encoding genes was detected by real-time PCR. The primer sequences are listed in Appendix A. Additionally, β-actin (GenBank ID: GU564232) gene was designed as an internal control. The same RNAs extracted for RNA sequencing were used in the real-time PCR experiment. The PCR amplification conditions were as follows: 50 °C for 3 min; 95 °C for 2 min; 39 cycles at 95 °C for 15 s and 60 °C for 30 s; 4 °C overnight. Relative fold changes were calculated based on the comparative cycle threshold (2^−ΔΔCt^) values.

### 2.11. Statistical Analysis

For enrichment analyses, a two-tailed Fisher’s text was applied to analyze the enrichments of the DAPs against all identified proteins. The threshold for false discovery rates (FDRs) of each protein under the NDRV infection was set at 0.01. Correction of multiple hypothesis testing was employed using the FDR analysis method. All enrichment terms with a *p* value < 0.01 were considered significant. Significant differences between different treatment groups were analyzed using a one-way analysis of variance with a Tukey’s test.

## 3. Results

### 3.1. Overview of the Proteomic Data

The dynamic changes in proteomes of the bursa of Fabricius under different conditions were quantified using an integrated TMT and LC-MS/MS approach (Figure 1a–e). PCA showed that the percentages of the PC1 and PC2 were 38.7% and 32.4%, respectively. Three groups were clearly separated, suggesting that there were great differences among different sample groups at the proteomic level (Figure 1d).

In total, 325,042 spectrums were detected, of which 83,907 spectrums were matched. Based on the matched spectrums, 49,505 peptides, including 44,129 unique peptides, were identified. Based on the unique peptides, 7075 proteins were identified, of which 5633 proteins were quantified (Figure 1e). Most peptides had 8–10 amino-acid residues, suggesting that the sampling process achieved the standards (Appendix A). The detailed information of all the identified peptides, including functional annotations, enrichments, and subcellular localizations is listed in Appendix A.

### 3.2. Analysis of the DAPs under HN10/JDm10 Infections

Among the quantified proteins, a number of DAPs were identified. In the HN10/control comparison, 210 DAPs, including 131 up- and 79 down-regulated proteins, were detected; in the JDm10/control comparison, 55 DAPs, including 29 up- and 26 down-regulated proteins, were identified; and in the HN10/JDm10 comparison, 198 DAPs, including 145 up- and 53 down-regulated proteins, were detected (Figure 2a). The numbers of the DAPs are shown using a Venn diagram (Figure 2b).

Under the HN10 infection, the highest up-regulated proteins were VWFD protein (5.69 folds), interferon-induced protein (4.42 folds), and UMP-CMP kinase 2 (4.16 folds), and the largest down-regulated proteins were pleiotrophin (0.37 folds), galectin (0.37 folds), and cysteine-rich secretory protein 3 (0.39 folds) (Appendix A). Under the JDm10 infection, the highest increased proteins were URB1 (1.91 folds), methylmalonyl-CoA epimerase (1.91 folds), and histone demethylase (1.80 folds), while the largest decreased proteins were component of oligomeric Golgi complex 2 (0.24 folds), mastermind-like proteins 3 (0.35 folds), and leiomodin-1 (0.56 folds) (Appendix A).

### 3.3. Classification of the DAPs under the HN10/JDm10 Infections

In the HN10/control comparison, the largest biological process GO terms were ‘cellular process’ (152 proteins) and ‘biological regulation’ (135 proteins); the largest cellular component GO terms were ‘cell’ (157 proteins) and ‘organelle’ (133 proteins); and the largest molecular function GO terms were ‘binding’ (116 proteins) and ‘catalytic activity’ (69 proteins). In the JDm10/control comparison, the largest biological process GO terms were ‘cellular process’ (35 proteins) and ‘biological regulation’ (32 proteins); the largest cellular component GO terms were ‘cell’ (37 proteins) and ‘organelle’ (18 proteins); and the largest molecular function GO terms were ‘binding’ (26 proteins) and ‘catalytic activity’ (12 proteins) (Figure 2c).

In the HN10/control comparison, 60 cytoplasm-, 49 extracellular-, and 39 nucleus-localized proteins were predicted (Appendix A); in the JDm10/control comparison, 16 cytoplasm-, 9 extracellular-, and 18 nucleus-localized proteins were predicted (Appendix A); and in the HN10/JDm10 comparison, 55 cytoplasm-, 42 extracellular-, and 43 nucleus-localized proteins were predicted (Appendix A).

### 3.4. Enrichment Analysis of the DAPs under HN10/JDm10 Infections

Under the HN10 infection were the up-regulated proteins enriched in ‘Neuroactive ligand-receptor interaction’ (Map04080), ‘Leishmaniasis’ (Map05140), and ‘Inflammatory bowel disease’ (Map0532); and the down-regulated protein enriched in ‘Melanoma’ (Map05218), ‘Proteoglycans in cancer’ (Map05205), and ‘Breast cancer’ (Map05224). Under the JDm10 infection were the up-regulated proteins enriched in ‘Platelet activation’ (Map04611), ‘Mineral absorption’ (Map04978), and ‘Longevity regulating pathway’ (Map04212); and the down-regulated protein enriched in ‘Autophagy’ (Map04140), ‘Non-alcoholic fatty liver disease’ (Map04932), and ‘Chemokine signaling pathway’ (Map04062) (Figure 3).

### 3.5. PPI Network Analysis of the DAPs under HN10/JDm10 Infections

PPI network analysis is a useful tool to predict the biological functions of the identified DAPs. Here, 33 DAPs in the HN10 vs. control comparison and 7 DAPs in the JDm10 vs. control comparison were considered as network nodes. In the HN10 vs. Control comparison, two greatly enriched interaction clusters, including the protease systems and the innate and adaptive immune system, were found in the networks (Appendix A).

### 3.6. Analysis of the DEGs under HN10/JDm10 Infections

Our transcriptomic analysis assembled 14,263 unigenes, among which a number of DEGs were identified. In the JDm10/Control comparison, 933 DEGs, including 830 up-regulated and 92 down-regulated genes, were identified; in the HN10/JDm10 comparison, 63 DEGs, including 31 up-regulated and 32 down-regulated genes, were identified; and in the HN10/Control comparison, 1724 DEGs, including 1464 up-regulated and 260 down-regulated genes, were identified (Figure 4a). The numbers of the DEGs are shown using a Venn diagram (Figure 4b).

In the JDm10/Control comparison, the DEGs were enriched in ‘Phagosome’ (ko04145), ‘ECM-receptor interaction’ (ko04512), ‘Prion diseases’ (ko05020), ‘Complement and coagulation cascades’ (ko04610), and ‘Cytokine–cytokine receptor interaction’ (ko04060) (Figure 4c); and in the HN10/Control comparison, the DEGs were enriched in ‘Complement and coagulation cascades’ (ko04610), ‘Cytokine–cytokine receptor interaction’ (ko04060), ‘Focal adhesion’ (ko04510), ‘Cell adhesion molecules’ (ko04514), and ‘Hippo signaling pathway’ (ko04392) (Figure 4d).

### 3.7. DAPs Related to the Serine Protease System and the Innate and Adaptive Immune Responses

The serine protease system consists of the complement part, the coagulation part, and the fibrinolytic part [27]. In detail, three coagulation factors, including coagulation factor II (R0LYC0), XIII (U3IBH9), and IX (R0JLH6), were detected; for the fibrinolytic system, three protein chains of fibrinogen, including α-chain, β-chain, and λ-chain, were identified; and for the complement system, 11 complement factors were identified (Appendix A).

The innate and adaptive immune responses play important roles in the regulation of host defenses [7]. In our study, 32 proteins related to the intracellular signaling pathways of PRRs were detected. In detail, four receptors, including one TLR receptor (TLR-3), and three RLR type receptors (RIG-I, MDA5, and LGP2); eight adaptors, including one TRAM, one TRIF, one MyD88, three IRAKs, and two TRAFs; 13 kinases, including eight MAPKs, three IKBK subunits, one TBK, and one RIP; and seven down-stream TFs, including six IRFs, and one NF-κB, were identified (Appendix A).

### 3.8. Comparison of the DAPs among Liver, Spleen, and Bursa of Fabricius Cells

The proteins responsive to the NDRV infection in the liver and spleen have been previously studied [17,18]. In the present study, proteins responsive to the NDRV infection in the bursa of Fabricius cells were identified (Table 1). In total, 16 proteins related to the serine protease system were detected in at least two of the three selected organs. Under NDRV infection, most of these proteins were up-regulated in the liver and spleen cells, and only eight of these proteins were up-regulated in the bursa of Fabricius cells. Interestingly, coagulation factor IX was up-regulated in the bursa of Fabricius, and not in the liver and spleen cells. All three chains of fibrinogen, including α-chain, β-chain, and λ-chain, were up-regulated in all three selected organ cells (Figure 5a).

A total of 29 proteins related to the intracellular signaling pathways of PRRs were detected in all three selected organ cells. Among these proteins, RIG-I, MDA5, and IRF3 were significantly up-regulated in all three selected organ cells. TLR-3 and IKBKB were only up-regulated in the liver cells, MAPK20 was only up-regulated in the bursa of Fabricius cells, and IRAK2 was only up-regulated in the spleen samples (Figure 5b).

### 3.9. Verification of the DAPs and DEGs under HN10/JDm10 Infections

Due to the lack of single sensitive biomarker, it is difficult to verify the changes in the proteins responsive to the HN10/JDm10 infections. In the present study, PRM assay was used to validate the differential accumulation levels of proteins responsive to the HN10/JDm10 infections. Seventeen proteins related to innate and adaptive immune responses were selected for the PRM verification. The relative accumulation levels of these selected proteins are presented in Appendix A. The trends of these DAPs tested by the PRM assay were agreed with the TMT-label quantification.

Furthermore, the expression levels of eight key genes were selected randomly and confirmed using qRT-PCR. The expression levels obtained from this assay were basically consistent with the RNA-seq results (Appendix A).

### 3.10. Integrated Proteomic and Transcriptomic Analysis Revealed the Immune Response to HN10/JDm10 Infections

For coagulation factors, coagulation factor IX was significantly increased under the HN10 infection. The encoding gene of coagulation factor IX was significantly up-regulated under both of the JDm10/HN10 infections. Three chains were significantly increased under the HN10 infection while α-chain and λ-chain of fibrinogen were up-regulated under the JDm10 infection. Most of the encoding genes of the fibrinogen λ-chain were significantly up-regulated under both of the JDm10/HN10 infections. For complement components, complement C1s, C5, and C8 were significantly increased under the JDm10 infection. Interestingly, most of the complement component encoding genes were significantly up-regulated under both of the JDm10/HN10 infections (Figure 6a). Under the HN10 infection, four innate and adaptive immune response-related proteins, including RIG-I, MDA5, MAPK20, and IRF3, were significantly up-regulated. No DAPs were detected under the JDm10 infection (Figure 6b).

The encoding gene of RIG-I was significantly up-regulated for both of the JDm10/HN10 infections. For the MDA5 encoding genes, DMA5_1 was significantly up-regulated for both of the JDm10/HN10 infections. No significant changes in the expression of DMA5_1 were observed. Several MAPK encoding genes, such as MAPK11, MAPK12, and MAPK13, were significantly up-regulated under HN10 infection and MAPK10 was significantly up-regulated under JDm10 infection. No IRF3 encoding gene was identified by transcriptome. Transcriptomic analysis showed that the encoding gene of IRF6 was significantly up-regulated under both of the JDm10/HN10 infections (Figure 7a,b).

## 4. Discussion

NDRV infection causes a high mortality rate for ducks and serious economic losses for the waterfowl industry [1]. For ducklings, disease caused by the NDRV infection is characterized by hemorrhagic-necrotic lesions in various visceral organs [28]. Exploration of the host responses to the NDRV infection is an essential way to improve the solutions for this lethal disease [29]. Our previous studies revealed the responses of liver and spleen to the NDRV infection at proteomic level. The bursa of Fabricius is a major organ that is affected by environmental antigens [14]. Exposure of the bursa of Fabricius to a wide variety of viral antigens leads to production of subsequent effective systemic responses [15]. However, the transcriptomic and proteomic responses of the bursa of Fabricius to the NDRV infection are largely unknown.

In poultry, several traditional 2D gel analyses of the bursa of Fabricius have been performed in the past years. For examples, 2D DGGE analysis of the bursa of Fabricius identified a subset of 153 differentially abundant proteins under different stages of chicken B-cell development [30]. Using the 2DE integrated MALDI-TOF MS method, 54 altered cell proteins were identified as differentially accumulated proteins in the bursa of Fabricius of chickens under the infectious bursal disease virus infection [31]. To analyze the host responses in the bursa of Fabricius of chickens under the Marek’s disease virus infection, 26 proteins were identified by the MALDI-TOF/TOF mass spectrometer method [32]. In our study, 7075 proteins were identified in the bursa of Fabricius, which was greater than in previous works. It suggested the need for a deeper exploration of the proteins that played roles in the immune responses of the bursa of Fabricius to the NDRV infections.

Recently, several transcriptomic analyses were performed to reveal the role of the bursa of Fabricius in poultry. Differential expression analysis showed that a number of immune-related genes, such as VEGFA, MyD88, IL15, and TLR4 in the bursa of Fabricius were significantly changed in a chicken stress model [33]. Comparison transcriptomic analysis revealed the differential expression of NOD-like and toll-like receptor signaling pathways-related genes in the bursa of Fabricius between Silky Fowl and White Leghorn [34]. In our study, a number of DEGs were identified under the JDm10/HN10 infections, suggesting that DRV infection also has an effect on gene expression in duck immune organs.

The classic DRV was first isolated in the 1970s in France and widely described in many other countries [32]. Recently, several novel varieties of DRV have been isolated from different organs of ducks, such as N-DRV-XT18 strain from the spleen of Pekin duck, NP03 strain from Muscovy duck embryo fibroblasts, and TH11 strain from Pekin ducklings [4,5,35,36]. However, there are few reports of natural attenuated strains of NDRV [21]. In the clinic, most NDRVs can cause obvious clinical pathological symptoms [4,5,36,37]. JDm10 is a laboratory made attenuated DRV [22]. In our study, the number of DAPs and DEGs under HN10 infection was larger than the JDm10 infection, confirming the virulent strain caused serious damage to bursa of Fabricius. In addition, the artificially attenuated JDm10 strain caused only weak damage to bursa of Fabricius. Interestingly, there are only 63 DEGs between the JDm10- and HN10-infected samples, while the number of DAPs is 198. Our data indicated that there are different regulation mechanisms between the gene transcription level and protein expression level.

PPI network analysis indicated that the DAPs under HN10 infection were enriched in the serine protease systems. The serine protease systems function in several immune reactions, such as damage repairing, pathogen removing, and anti-virus [27,38]. In the bursa of Fabricius, most of the serine protease system-related genes were significantly up-regulated under both of the JDm10/HN10 infections, suggesting the activation of the serine protease system. However, these serine protease system-related gene encoding proteins were only significantly up-regulated under HN10 infection. The serine protease system consists of three major autonomous proteolytic cascade parts, including the complement, coagulation, and fibrinolytic parts [39]. Previous studies revealed that the coagulation cascade system can be activated by the invasion of pathogenic microbes [40]. Fibrinogen, consisting of three different disulfide-linked polypeptide chains, is a soluble 340-kDa protein [41]. In duck, three fibrinogen chains were increased after HN10 infection, indicating a higher concentration of fibrinogen in the blood of HN10 infected ducks than in the JDm10 infected ducks. Taken together, our data suggested that the serine protease systems at protein level were not obviously activated under the JDm10 infection in ducks. On the other hand, it also shows that the virulence of the JDm10 strain has been completely attenuated by artificial passage and has no pathogenicity to the host, so it can be used as a vaccine candidate strain for NDRV attenuated vaccine.

The host innate immune system recognizes invasive microorganisms and counters against their stimuli [42]. RIG-I and MDA5 are essential for the recognition of RNA viruses [43]. In our study, both *RIG-I* and *MDA5_1* genes were significantly up-regulated by DRV infection, suggesting that the DRV strain was quickly recognized by the innate immune system. However, the RIG-I and MDA5 proteins are only significantly accumulated under HN10 infection. Then, the signals are delivered to kinases, such as MAPKs, to activate several downstream transcription factors, such as IRFs. In our study, MAPK20 and IRF3 were significantly up-regulated by HN10 infection, suggesting their important roles in receptor signal transduction.

The disease causing novel duck reovirus is mainly from hemorrhagic-necrotic lesions in the visceral organs [3,44,45]. The bursa of Fabricius is a lymphoid organ involved in destroying pathogens [46]. However, the distinct innate immune responses to NDRV infection in different visceral organs are largely unknown. In ducks, the liver and spleen were the organs that seriously suffered from NDRV causing disease, covered with hemorrhage and necrotic lesions [17,18]. Under the NDRV infection, accumulation levels of the serine protease system-related proteins in liver and spleen were higher than in the bursa of Fabricius, suggesting that liver and spleen function as the front-line organs of host defense against viral infection, and the serine protease system plays a significant role in host resistance to NDRV infection. Coagulation factor IX is an essential constituent of the coagulation cascade, which serves as a target for antiviral treatment [47]. In our study, coagulation factor IX was significantly up-regulated in the bursa of Fabricius, not in the liver and spleen samples, indicating the potential roles of the bursa of Fabricius in antivirus.

## 5. Conclusions

In summary, many NDRV infection responsive proteins in the bursa of Fabricius of C. moschata were detected by integrated transcriptomic and proteomic analysis. Most of the immune-related genes were up-regulated under both JDm10/HN10 infections. However, the immune-related proteins were only accumulated under HN10 infection. Coagulation factor IX of the coagulation system was significantly increased in the bursa of Fabricius, not in the liver and spleen samples. However, most DAPs in the liver, spleen, and bursa of Fabricius were enriched in different factors of the serine protease systems, suggesting a significant role of serine protease systems in the host’s resistance to NDRV infection. The results will provide a valuable resource for revealing the regulation mechanism related to the responses of the bursa of Fabricius to NDRV infections.

## Figures and Tables

**Figure 1 viruses-14-01615-f001:**
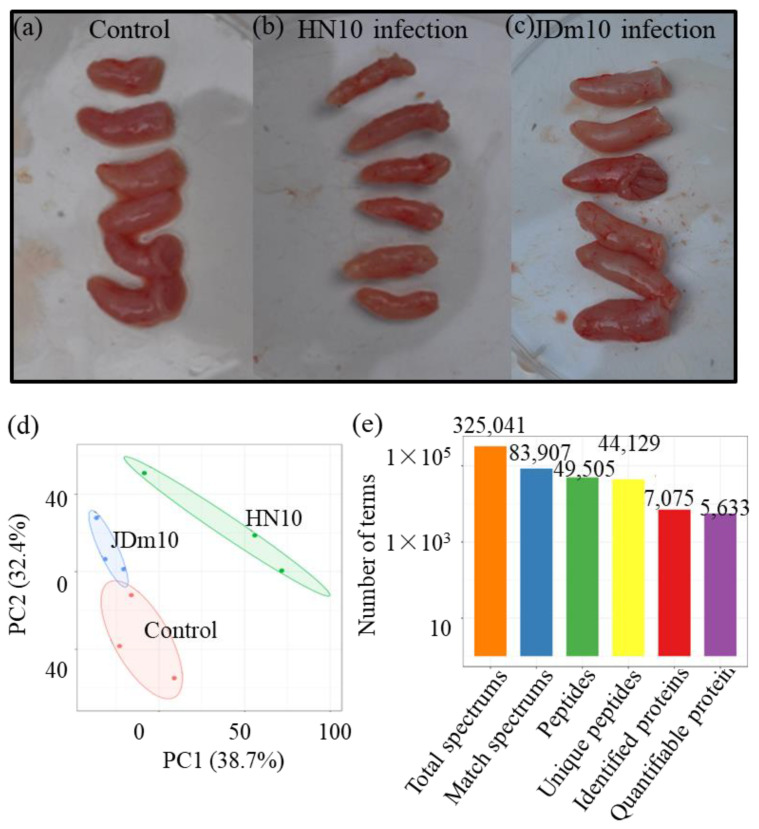
The basic information of the proteomes from different samples. (**a**–**c**) The pictures showed the bursa of Fabricius under the control and two NDRV strain infections. (**d**) The PCA data of the samples from the three different groups, including the control, HN10 infection, and JDm10 infection. (**e**) The number of terms in each category.

**Figure 2 viruses-14-01615-f002:**
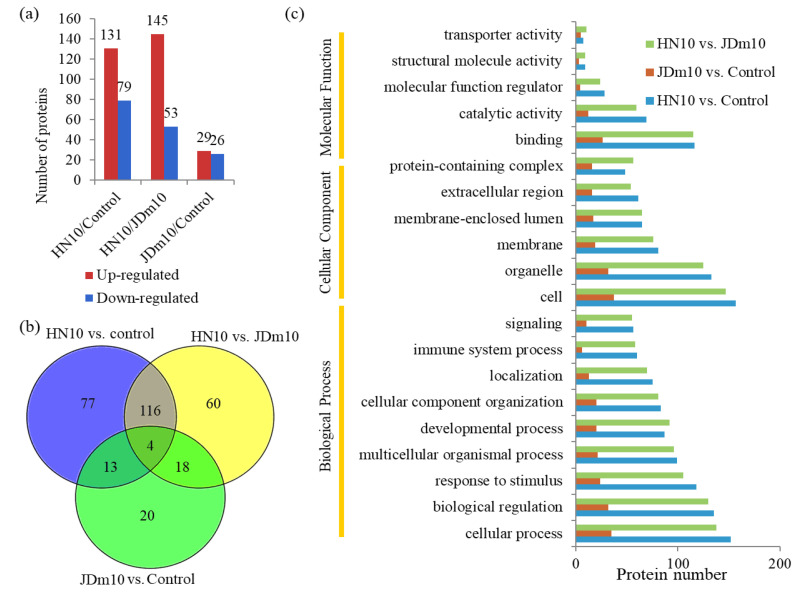
Analysis of the DAPs under the HN10 and JDm10 infections. (**a**) The numbers of the up- and down-regulated protein in different comparisons, including the HN10/control, JDm10/control, and HN10/JDm10 comparisons. (**b**) Venn diagram showing the numbers of DAPs in different comparisons. (**c**) GO analysis of the DAPs in different comparisons. All the DAPs were classified by their GO terms belonging to cellular component, mole function, and biological process.

**Figure 3 viruses-14-01615-f003:**
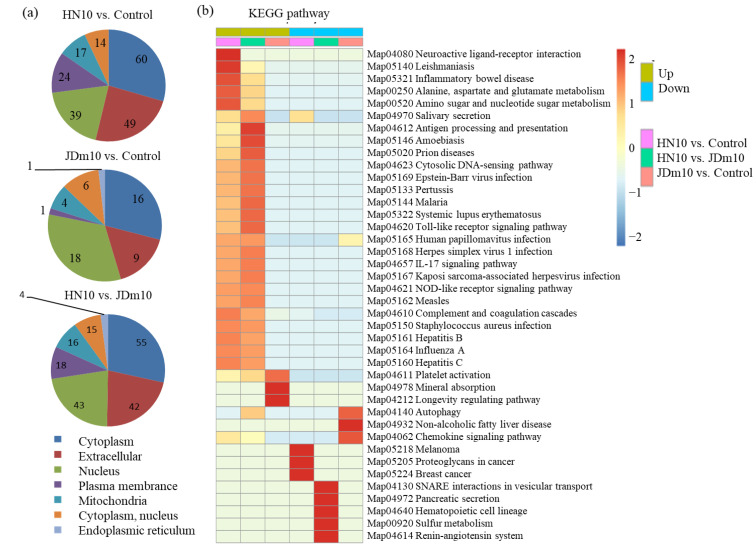
Subcellular locations and KEGG enrichment analysis of the DAPs under the HN10 and JDm10 infections. (**a**) Subcellular locations of the DAPs in different comparisons, including JDm10/control, HN10/JDm10, and HN10/control groups. (**b**) KEGG enrichment analysis of the DAPs in different comparisons, including JDm10/control, HN10/JDm10, and HN10/control groups.

**Figure 4 viruses-14-01615-f004:**
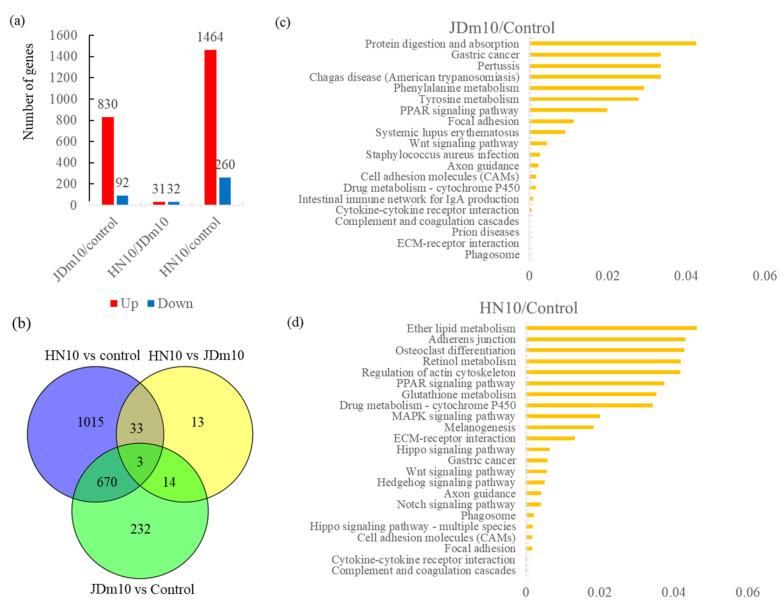
Analysis of the DEGs under HN10/JDm10 infections. (**a**) Number of DEGs in different comparisons, including JDm10/control, HN10/JDm10, and HN10/control groups. (**b**) Venn diagram showing the number of DEGs in different comparisons. (**c**) KEGG enrichment analysis of the DEGs under JDm10/Control comparison. (**d**) KEGG enrichment analysis of the DEGs under HN10/Control comparison.

**Figure 5 viruses-14-01615-f005:**
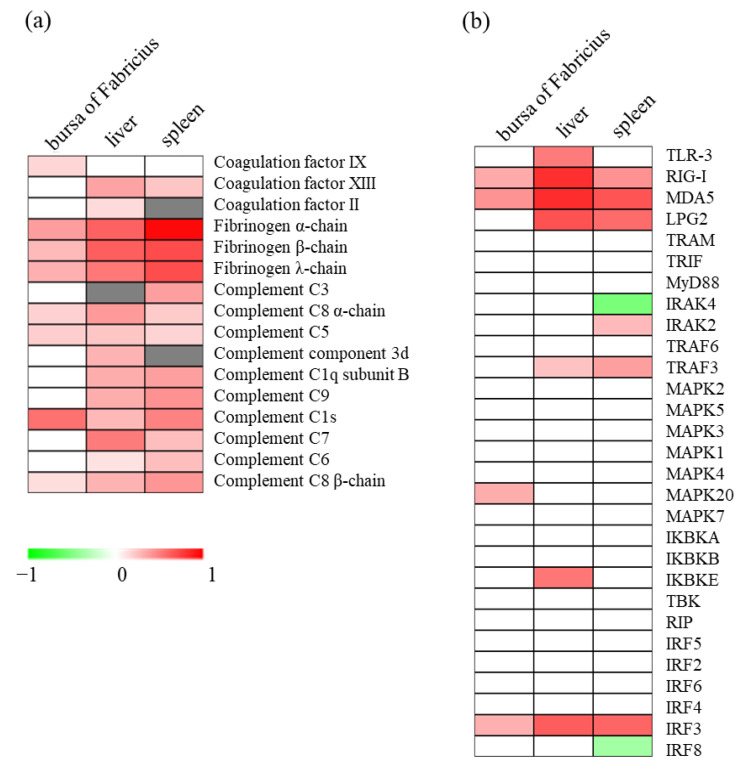
Comparison of the DAPs among liver, spleen, and the bursa of Fabricius cells. (**a**) Serine protease system-related proteins among different organs. (**b**) Intracellular signaling pathway-related proteins among different organs. Red indicates up-regulated proteins, green indicates down-regulated proteins, and grey indicates un-quantified proteins.

**Figure 6 viruses-14-01615-f006:**
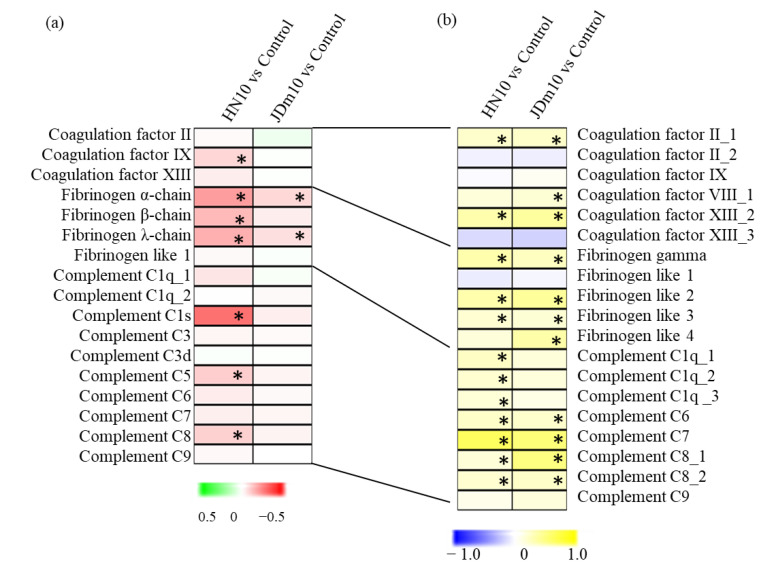
Integrated proteomic and transcriptomic analysis of the serine protease system-related proteins and genes. (**a**) Differential accumulation of serine protease-related proteins. The heatmap scale ranges from −0.5 to +0.5 on a log2 scale. (**b**) Differential expression of serine protease-related genes. The heatmap scale ranges from −2 to +2 on a log2 scale. “*” indicates significant differences in each comparison (*p* < 0.05).

**Figure 7 viruses-14-01615-f007:**
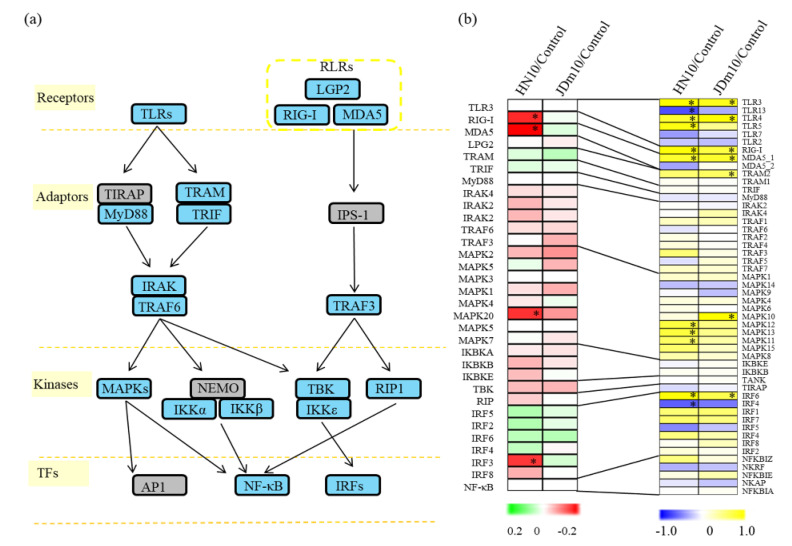
Involvement of the innate and adaptive immune system in response to the HN10/JDm10 infections. (**a**) Overview of the innate and adaptive system in Muscovy duckling. Blue backgrounds indicate the identified proteins. Grey backgrounds indicate unidentified proteins. (**b**) The accumulation profiles of the proteins involved in innate and adaptive immune system under the HN10 and JDm10 infections. The heatmap scale ranges from −2 to +2 on a log2 scale. “*” indicate significant differences in each comparison (*p* < 0.05). The heatmap was drawn using R Package heatmap (v.2.0.3 https://cran.r-project.org/web/packages/cluster/, accessed on 28 March 2022).

**Table 1 viruses-14-01615-t001:** DAPs involved in the innate and adaptive immune responses.

Protein Accession	Protein Description	Bursa of Fabricius	Liver	Spleen
R0JLH6	Coagulation factor IX	1.5	1.0	0.9
U3IBH9	Coagulation factor XIII	1.2	2.3	1.7
R0LYC0	Coagulation factor II	1.1	1.4	NA
R0JSX9	Fibrinogen α-chain	2.4	4.2	9.3
U3I9E6	Fibrinogen β-chain	1.9	4.3	5.1
U3IA23	Fibrinogen λ-chain	2.1	3.4	5.0
U3J6P0	Complement C3	1.1	NA	2.4
U3IKF3	Complement C8 alpha chain	1.5	2.5	1.6
R0JIF4	Complement C5	1.6	1.7	1.5
B5AG23	Complement component 3d	1.0	2.0	NA
R0LW73	Complement C1q subunit B	1.3	2.1	2.4
U3IN55	Complement C9	1.1	2.1	2.7
U3ID07	Complement C1s	3.6	1.9	3.1
U3INH5	Complement C7	1.2	3.3	1.8
U3IQV8	Complement C6	1.2	1.3	1.8
U3IKF3	Complement C8 beta chain	1.4	2.0	2.6

## Data Availability

The saw MS/MS data were deposited in the proteome Exchange Consortium by the PRIDE partner repository with ID PXD025093.

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
