# Peer review of "Intergrated Transcriptomic and Proteomic Analysis Revealed the Differential Responses to Novel Duck Reovirus Infection in the Bursa of Fabricius of *Cairna moschata"

_viruses, 2022, doi:10.3390/v14081615_

Round 1

Reviewer 1 Report

Dear Authors:

These comments on the research are as follows.

1.     Suggestion on the abstract where present some data related to antiviral immunity based on the research result.

2.     A brief description of the differences in clinical symptoms caused by the two strains in Introduction.

3.     Lines 78-82, the background of the two strains need to be supplemented. If the authors have been published papers added the references, or sequenced the complete genome added the Genbank NO.

4.     Line 85, the two words of 108.2 and 106.4 have formal errors.

5.     Lines 399-405, Is the description on two strains of JDm10 and HN10 supported by data or published in the public literature

Reviewer 2 Report

The authors analyzed the transcriptomes and proteomes of the samples from control and two NDRV strain (HN10 and JDm10) with different virulence. A number of differentially expressed genes and differential accumulated proteins were found too be enriched in the serine protease system and innate immune response clusters. The work provided a detailed resource for studying the proteins participated in the resistances of the bursa of Fabricius of duck to the NDRV infections. 

There are several comments:

1. Several previous studies have focused on the responses of ducks to DRV infection. In the introduction section, all previous works should be mentioned. 

2. The function of the bursa of Fabricius in duck should be brief introducted.

3. Three organs, including the bursa of Fabricius, liver and spleen, were compared. The response of the bursa of Fabricius to DRV is very weak. Why?

4. More disscusion should be added to the Disscusion section. 
